# Hotspot Analysis of Rumen Microbiota and Methane Mitigation in Ruminants: A Bibliometric Analysis from 1998 to 2023

**DOI:** 10.3390/ani15050681

**Published:** 2025-02-26

**Authors:** Xueyue Zheng, Lijie Tang, Rong Wang, Xiumin Zhang, Min Wang, Duanqin Wu

**Affiliations:** 1College of Animal Science and Technology, Hunan Agricultural University, Changsha 410128, China; zhengxueyue1111@163.com (X.Z.); 19191018407@163.com (L.T.); 2CAS Key Laboratory for Agro-Ecological Processes in Subtropical Region, National Engineering Laboratory for Pollution Control and Waste Utilization in Livestock and Poultry Production, Institute of Subtropical Agriculture, the Chinese Academy of Sciences, Changsha 410125, China; xmzhang@isa.ac.cn (X.Z.); mwang@isa.ac.cn (M.W.); 3Institute of Bast Fiber Crops, Chinese Academy of Agricultural Sciences, Changsha 410205, China

**Keywords:** greenhouse gas, methane emission, rumen microbiota, bibliometrics

## Abstract

Enteric methane emissions from livestock are a significant source of anthropogenic greenhouse gases and a contributor to climate change. A significant portion of this methane is derived from the digestive activities of ruminants. The present study determined research trends on rumen microbiota and methane mitigation from livestock over the last two decades. This study evaluated 189 articles published between 1998 and 2023, identifying hotspots and research gaps in the field of enteric methane mitigation. The number of studies increased exponentially from 2008, with most studies being conducted in China and North America, where livestock farming is widespread. The data revealed that although some progress has been made, further research is still needed, particularly on sustainable and effective CH_4_ mitigation strategies. This information can be used to reduce the environmental impacts of livestock production, contributing to sustainable food production systems and climate change mitigation.

## 1. Introduction

Methane (CH_4_) is the second-most abundant greenhouse gas, following carbon dioxide (CO_2_), and has a warming potential 28 times greater than CO_2_. It contributes up to 25% to global warming [1,2]. The Intergovernmental Panel on Climate Change (IPCC) has consistently emphasized the significance of CH_4_ mitigation in decelerating global warming, and issued the Kyoto Protocol in 1997, which formally took effect from February 2005. This marked a historic milestone, as it was the first instance where emissions were regulated to limit their impact on climate change [3]. Consequently, in November 2015, the IPCC introduced the Paris Agreement, which stands as a significant milestone in international legal frameworks after the Kyoto Protocol. The Paris Agreement set out detailed global actions to combat climate change and stipulated that the world should achieve net-zero greenhouse gas emissions in the second half of this century. The goal of the Paris Agreement is to reduce the ecological risks of climate change and prevent climate change from posing a potentially existential threat to humankind [4,5].

Atmospheric CH_4_ emissions come from natural ecosystems and human activities [6,7], of which livestock emissions contribute about 33% to the total emissions and are the largest single agricultural source [8]. Cattle and sheep are the main contributors (80%) to all livestock CH_4_ emissions [9]. About 90% of CH_4_ is produced in the rumen, and thus the reduction of ruminal CH_4_ production is a key approach to CH_4_ mitigation [10]. CH_4_ emissions are also considered a major pathway of feed energy loss (about 2% to 12% of the total digestible energy) [8,11]. Therefore, studies on the reduction of ruminal CH_4_ emissions can potentially bring benefits to both alleviating greenhouse effect and improving feed energy utilization efficiency [12].

Nutritional strategies for reducing CH_4_ emissions in ruminants include dietary composition, feed quality improvement, rumen passage rate increase, oil, and CH_4_ inhibitors [11,13,14]. It has been shown that a reduction in methanogen copy number and the suppression of ruminal hydrogen in CH_4_ synthesis can effectively reduce CH_4_ emissions [15] and that changes in rumen microbial community structure and fermentation patterns can reduce hydrogen production, thus reducing CH_4_ emissions [16]. These studies show that changes in rumen microbiota are associated with a reduction in CH_4_ emissions. Although many authors have reported on the methods, advantages, and disadvantages of CH_4_ mitigation in ruminants [17,18,19], there is little literature on the progress and future research directions of CH_4_ mitigation related to rumen microbiota. Therefore, this paper analyzes the current research status on rumen microbiota in relation to CH_4_ emissions by comparing journals and evaluates the international cooperation and research focus in this field. The purpose of this study was to understand the current research progress and the role of rumen microbiota in CH_4_ mitigation and to provide scientific guidance for future research in this field.

## 2. Materials and Methods

### 2.1. Data Sources and Research Strategies

Web of Science (WoS) is a multidisciplinary literature database developed by Thomson Reuters Corporation within the open Internet environment. It contains the most significant scientific search engines and pertinent bibliometric data retrieval databases. It is widely acknowledged as the foremost quality-oriented database globally, offering a standardized record for retrieving global scientific literature across diverse research fields [16]. WoS provides bibliometric and network analysis, with important metadata such as abstract, type of document, timeline, citation, list of scholars, country information, university affiliation, and journal impact factor. WoS is an open access database that does not involve human subjects and does not require institutional review board approval [20,21]. This study utilizes the WoS database as its primary source of scientific data. The databases employed encompass various indices, such as the Science Citation Index Expanded (Expanded), Social Sciences Citation Index (SSCI), Emerging Sources Citation Index (ESCI), Conference Proceedings Citation Index-Science (CPCI-S), Book Citation Index—Social Sciences and Humanities (BKCI-SSH), and Arts and Humanities Citation Index.

Advanced search was used to find publications related to rumen microbiota and CH_4_ emissions in ruminants. Selection of keywords is very important in analysis, as the choice of keywords directly affects the results of a study. Therefore, we used different combinations of keywords in several cycles [22]. Finally, we used the following retrieval strategy to find appropriate search terms using Boolean logic (i.e., OR and AND): TS  =  (“microorganism” OR “germ” OR “microbe” OR “Microbiology” OR “Microbial”) AND (“cow” OR “Cattle” OR “Bovine” OR “steer” OR “sheep” OR “goats” OR “cud chewer” OR “ruminant” or “ruminates” or “OX” or “bull” or “heifer” or “buffalo”) AND (“methane” or “CH_4_”). The retrieval period was from 1 January 1998 to 16 December 2023. The literature search resulted in a total of 1638 references after screening for articles written in English within the specified time frame. Among these, we found 191 studies related to rumen microbes and CH_4_ emissions based on a review of the title and abstract. Then, according to the data source of the article, the bibliometric scoring method was used to conduct quantitative and qualitative analysis of the development trends and patterns of research in this field (Figure 1).

### 2.2. Statistical Analysis

This study categorized the relevant research into three distinct phases, each corresponding to significant international policies concerning CH_4_ emissions from ruminants: January 1998–January 2005 (issuance of the Kyoto Protocol), February–October 2015 [23], and November 2015–December 2023 (proposed Paris Agreement). These phases represent the primary clusters within the trend of research on rumen microbiota and CH_4_ mitigation. An analysis of shifts in research trends within this field has been conducted, encompassing a comprehensive examination of impact factors and citation rates associated with relevant journals. The assessment of research in this study included: (1) publication output and subject category according to the WoS database (from 1998 to 2023); (2) the most influential among the top ten journals; (3) the most cited articles; (4) the research productivity of the top ten countries; and (5) screening of field keywords. Additionally, impact factors, derived from an average number of citations from Journal Citation Reports (JCR) published in 2022, were utilized to evaluate journal performance via SCIMago Journal Rank (SJR).

Publication trends from 1998 to 2023 were assessed using Excel 2019, with a focus on publication volumes, and bibliometrics was subsequently employed to examine the existing number of publications, citation frequency, and author distribution associated with the rumen microbiota and methane mitigation, in order to discern current research trends. Furthermore, the assessment of citation frequency and impact factor contributed to an objective foundation for academic evaluation. Author credibility was determined through the evaluation of total citations (TCs) and average citations per paper (TCs/P), while TCs were also used as a criterion for gauging the reputation of the source institution. Social network analysis was subsequently employed to discern dynamic patterns in research focus and collaborations among leading authors, countries, and institutions. Both VOSviewer 1.6.20 and CiteSpace 6.2.R6 software enabled quantitative analysis and visualization of literature units and keywords [24,25], with “Country” depicting institutional cooperation networks nationally and “Keyword” representing high-frequency keyword co-occurrence map analysis [26]. Additionally, co-citation analysis was leveraged to explore the interconnections between cited documents and the evolution of study content and dynamics [27].

## 3. Results and Discussion

### 3.1. Number of Published Articles

#### 3.1.1. Publication Output

The present study analyzed a total of 191 papers, which were contributed by 889 authors affiliated with 313 organizations from 46 countries. These publications appeared in 63 journals and cited a total of 6012 papers from 1767 distinct journals. Figure 2 depicts the publication trends concerning rumen microbiota and CH_4_ mitigation. During the initial phase (1998–2005), the annual number of publications on these selected topics did not exceed two, resulting in a cumulative total of four publications. This suggests that research in this domain was still at an early stage, possibly due to the recent inception of the Kyoto Protocol. Although nations began addressing concerns related to CH_4_ emissions during this period, no relevant policies were introduced. In the subsequent phase (2006–2015), there was a significant surge in publications focusing on these subjects, reaching a count of 134 papers, which constituted approximately 70.9% of the overall output. The citation counts also experienced substantial growth, amounting to approximately 5500 citations. This increase can be attributed to heightened scholarly interest following the implementation of the Kyoto Protocol. In the final stage (2016–2023), although there was no further increase in publication numbers indicating sustained focus on research topics, the momentum seemed to have stabilized compared to previous years. This stabilization could be attributed to an evolution over a decade within dedicated teams researching rumen microbiota and CH_4_ mitigation. However, the annual citation counts continued to decline, possibly because more articles were available for citation and researchers had more options when choosing which articles to cite.

#### 3.1.2. Number of Articles Published by Journals

The analysis of the literature sources revealed that apart from a few journals in veterinary and food science technology, many publications over the past 26 years are primarily associated with animal science and microbiology. This concentration is evident in terms of publication volume. Specifically, four journals have published more than 10 papers each—*Journal of Dairy Science* (impact factor: 3.5), *Frontiers in Microbiology* (impact factor: 5.2), *Journal of Animal Science* (impact factor: 3.3), and *Animal Feed Science and Technology* (impact factor: 3.2). These journals focus on key areas such as nutrition, physiology, feed science, microbiology and animal production, predominantly targeting beef cattle, dairy cows, sheep, and other livestock and poultry species (Table 1). According to Bradford’s law in bibliometrics [28,29], a certain field’s journal can be divided into three parts, where one-third of articles concentrate on a few specific journals—consistent with this study’s findings. Through analyzing the citation patterns among these journals, it was found that the *Journal of Dairy Science* received the highest number of citations per article—an average of 53.42. This prominence is likely due to its reputation as a leading journal in the field of animal science, attracting considerable industry attention. Furthermore, it may be attributed to the fact that the dairy industry bears the brunt of CH_4_ emissions originating from livestock. The second most frequently cited journal was *Frontiers in Microbiology*, which had an average citation rate of 32.26 per article. This may be attributed to the journal’s high impact factor and its open access policy.

#### 3.1.3. Most Cited Research Article

The SCI mainly uses citation count as a key indicator to measure the quality and influence of scientific papers [30,31]. Here, the 191 papers had a total of 6349 citations, with an average citation frequency of 33.24 times per paper. The 10 most cited papers on enteric CH_4_ emission regulation by rumen microbiota during the last 26 years are summarized in Table 2. The paper “Specific microbiome-dependent mechanisms underlie the energy harvest efficiency of ruminants” by Ben Shabat SK et al. (2016) had the highest citation frequency (409 times) and the highest annual citation frequency (51.13 times/year) [32]. In this study, the taxonomic composition, gene content, microbial activity, and metabolite composition of rumen microbiota were characterized through diversity + metagenomics + metabolomics technology in 146 dairy cows with different feed conversion rates. The potential mechanism of different feed conversion rates from various angles was explored, and the results showed that the complex relationship between microbial genes, low species richness, and high feed conversion rate phenotype can accurately predict animal feed conversion rates. This study indicated that the enrichment of specific microorganisms and their metabolic pathways could enhance the utilization of energy and carbon sources in animals and reduce CH_4_ emissions. This is of great significance for the rational use of feed resources by rumen microbiota to improve the ecological environment of livestock farming.

Five of the ten most-cited articles were published in 2014–2018, suggesting that this period was particularly important for the development of the field. Interestingly, half of the most-cited papers were not published in journals with a high number of publications, and they came from biology journals more frequently than animal science journals. This is likely because impact factors are generally higher in biology journals, and many have interdisciplinary sections. Another possible explanation is that researchers prefer to publish their work in high-impact interdisciplinary journals to accelerate the field’s progress.

### 3.2. Analysis of Authors’ Countries

#### 3.2.1. Number of Articles by Country

As shown in Figure 3, the literature involving rumen microbial and CH_4_ mitigation originates from 46 countries around the world. The figure shows the number of published papers by country or region. China (42), the United States (23), Thailand (23), Canada (21), France (15), Australia (14), New Zealand (12), Spain (12), the United Kingdom (11), and Switzerland (9) are the 10 countries with the most publications in the field of rumen microbial and CH_4_ mitigation over the past 23 years. The number of papers published by scholars in China accounts for 21.99% of all papers on this topic, indicating that China has carried out more research in this field than any other country. This is related to China’s long-term development of carbon reduction regulations and policies in animal husbandry. For example, since 2001, China has issued carbon reduction policies for animal husbandry, including The Animal Husbandry Law of the People’s Republic of China, The Standard for Pollutant Emissions from Livestock Farming Industry, and so on. In 2007, the State Council issued Opinions on Promoting Sustainable and Healthy Development of Animal Husbandry, which was the first time that the State Council put forward the requirements of carbon neutrality in animal husbandry in the form of normative documents during China’s reform and opening up. Since 2015, the Animal Husbandry Law of the People’s Republic of China and the Environmental Protection Law of the People’s Republic of China have been formally implemented one after another, further forming a complete legal system of animal husbandry and providing legal guarantees for the sustainable development of animal husbandry. The number of policy texts concerning carbon reduction in animal husbandry has increased significantly, including training, subsidies, pilot projects, assessments, and other aspects.

#### 3.2.2. Co-Authorship Among Countries

Figure 4 illustrates the extensive collaboration among national authors in the field of rumen microbial and CH_4_ mitigation, with 46 nations contributing to published papers and a total of 148 links established. Notably, China emerges as a prominent player in this research area, exhibiting the highest density of links (with a total link strength of 28). This prominence can be attributed to President Xi’s initiatives since the 18th National Congress of the Communist Party of China, aiming to foster international cooperation for building a beautiful Earth and a community with shared future for mankind through One Belt and One Road, as well as accelerating ecological civilization reform domestically to create a beautiful China. Through effective diplomatic efforts, China has actively engaged in global initiatives focused on reducing CH_4_ emissions. Numerous domestic universities and research institutions have initiated international cooperation platforms, such as the International Research Center on Dairy Science at the Institute of Animal Sciences within the Chinese Academy of Agricultural Sciences. These endeavors have progressively positioned China as an indispensable partner within the international scientific research cooperation network.

International cooperation, particularly that facilitated by the United States (which has 17 links with a total link strength of 25), is marginally surpassed by China’s efforts. This is followed by Canada (with 11 links and a total link strength of 24) and Australia (which also has 11 links, but with a total link strength of 16). In relation to temporal development, countries including Canada, the United States, France, Australia, and the Netherlands commenced research on rumen microbial and CH_4_ mitigation comparatively early, around 2015. More recently, India and Costa Rica have entered this field within the last three years (2021–2023). The Global Inventory Report, released by the United Nations, confirms that all countries’ gas emissions are significantly deviating from the targets set by the Paris Agreement. Consequently, it is incumbent upon all nations to perpetuate their efforts towards system transformation to integrate carbon reduction into every facet of human life.

### 3.3. Research Hotspots and Emerging Trends Based on Keyword Co-Occurrence Analysis

#### 3.3.1. Research Hotspot Trends of Change

Keywords can clearly show the research status and development trends of scholarly articles. The Network Visualization view of VOSviewer 1.6.20 was used for keyword co-occurrence analysis, and keywords with co-occurrences greater than five were filtered to construct a network diagram of research hotspot changes (Figure 5). Fewer keywords were found in the early literature, which was basically after 2010, and was consistent with the distribution of publication volumes. Before 2015, the research focused on feed additives such as coconut oil [41,42], condensed tannins [43,44] and monensin [45]. Since 2020, microbiota (microbial synthesis, 16s metabarcoding) have gradually increased, indicating that the research focus has changed from macronutrients to molecular nutrition [46,47]. In addition, an increasing number of studies have begun to explore the regulatory effects of rumen microbiota on CH_4_ emissions.

The co-occurrence analysis showed that besides the keywords CH_4_, methanogenesis, rumen, and ruminant (dairy cows, sheep, and cattle), the frequency of rumen fermentation (or ruminal fermentation) and diversity also increased (Figure 5). This indicates that more and more researchers studying the effect of rumen microbiota on CH_4_ emission in ruminants have focused on the indicators related to rumen fermentation. The main reason is that the VFAs produced by rumen fermentation are often accompanied by hydrogen production, which is the main precursor of ruminal methanogenesis and closely related to CH_4_ emissions [48,49]. More and more in vivo experiments were conducted to study the effects of changes in rumen microbial diversity on CH_4_ emissions.

#### 3.3.2. Keyword Cluster Analysis

Keyword clustering analysis is a process that distills the co-occurrence network relationship into a more manageable number of clusters, relying on cluster statistics. In this study, we employed CiteSpace 6.2.R6 software to systematically conduct keyword clustering, enabling us to discern the relationships and primary research areas across three distinct stages (1998–2005, 2006–2015, and 2016–2023). It is noteworthy that the recurring presence of certain keywords underscores the significance of vocabulary in the current research phase.

In the keyword network map of phase I (1998–2005), two clusters were found (Figure 6A). In this phase, microbial protein synthesis was the focus, as reviewed by [50,51]. The studies used different feed additions (e.g., lotus corniculatus) or feed treatments and studied their effect on CH_4_ emissions [52]. Most studies in this first phase were performed with traditional animal nutrition. This was due to a lack of technology at that time and knowledge of background mechanisms, which led to research on a more macro scale.

In phase II (2006–2015), a keyword network map was generated, resulting in the identification of eight distinct clusters (Figure 6B). This contrasts with phase I, where novel feed additives such as condensed tannins and eucalyptus leaf meal were introduced [53,54]. Steers were predominantly used as the experimental animal. Metatranscriptome analysis emerged as a significant technique for measurement purposes. The research focus during this phase was primarily on methanogens, microbial community dynamics, and synthesis processes [55]. These results indicated that the regulation of ruminant feed was still the main research focus for controlling CH_4_ emissions in this field.

In phase III (2016–2023), eight clusters were obtained from the keyword network map (Figure 6C). In contrast to the previous two stages, the research in this stage mainly used steers and dairy cows as subjects, and the metatranscriptome and 16S sequencing continued to be the main research methods [56]. At the same time, new research content such as feed intake, heritability, and dissolved hydrogen began to appear [57]. The increase in the number of research topics and the cross-disciplinary characteristics are consistent with the current research trends. It can also be seen that in recent years, research on the mechanisms of CH_4_ generation has been gradually strengthened, and the reduction in CH_4_ emissions at the source is the focus of current research, which is further developed on the basis of the previous two stages [58]. Based on the three phases, it can be seen that scholars have paid more attention to the regulation of feed additives on rumen microbiota and CH_4_ emissions in ruminants over the past 26 years (Figure 6D). In addition to CH_4_ emission, feed intake, digestibility, and microflora were also important when choosing research content [59].

#### 3.3.3. Generation and Analysis of Burst Word

Burst words are keywords with a high frequency in a given period. They reflect the latest research hotspots and the development of theories and subjects at the forefront of research. CiteSpace’s burst detection function was used to analyze the high-frequency words that have appeared in recent years so as to further understand the research hotspots in this field. Figure 7 show several obvious periods of burst words. The use of microbial protein synthesis increased significantly from 1999 to 2008. After 2008, focus turned to the study of feed additives such as coconut oil, and then gradually turned to fiber, metabolism, enteric CH_4_, supplementation, methanogenic archaea, and so on [60,61]. In recent years, the dynamics of CH_4_ production, methods of evaluating nutrient digestibility, and ruminal fermentation parameters are also important topics.

The initial phase of research in ruminant nutrition primarily emphasized the enhancement of rumen microbial protein production [62]. Given that microbial proteins can supply 70% to 100% of the amino acids necessary for ruminants, the augmentation of microbial protein production could decrease the requirement for additional crude protein supplementation, thereby reducing feeding costs [63]. As a result, coconut oil has become a new field of research interest due to its cost-effectiveness, palatability, and accessibility as a feed resource. Furthermore, coconut oil is abundant in saturated medium- and long-chain fatty acids, which have the potential to diminish the ruminal protozoa population [64]. This reduction could boost rumen microbial protein synthesis, offset potential deficiencies in microbial protein availability, and inhibit the growth of rumen methanogens, consequently decreasing CH_4_ generation [65]. The emergence of terms such as “rumen fermentation”, “metabolism”, and “methanogenic archaea” predominantly occurred between 2015 and 2020, indicating a shift in research focus towards unraveling the pathway of CH_4_ generation. Methanogens play a crucial role within this context. This trend also suggests a transition in research content towards the microbial domain.

## 4. Future Research Directions

The increasing focus on CH_4_ emissions from ruminant livestock has led to a surge in research exploring the relationship between ruminal microorganisms in ruminants and CH_4_ emissions [66,67]. Most current studies in this field concentrate on short-term animal feeding experiments (lasting 21 to 42 days) employing a variety of direct measurement methods, such as respiration chambers, sulfur hexafluoride tracing technology, and the GreenFeed system to assess the impact on emission reduction. However, there is a relative paucity of research on long-term strategies. Additionally, the use of significantly different measurement methods across various institutions introduces a degree of uncertainty and inconsistency in evaluating CH_4_ emission reduction effects.

To address these limitations, future research needs to delve more deeply into the disparities between various measurement techniques to enhance measurement precision and data trustworthiness. Concurrently, it is imperative to amalgamate current research insights to scrutinize the temporal and spatial dynamics of CH_4_ emissions. This would bolster the regional CH_4_ emission dataset and furnish theoretical and empirical backing for the development of effective emission reduction strategies. As research attention transitions from macro-phenotypes to rumen microorganisms and their CH_4_ production pathways, there will be a growing emphasis on studies that employ additives for targeted regulation to curtail CH_4_ emissions. This approach is poised to become a dominant research trend, thereby elevating the efficiency of CH_4_ emission reduction.

Nonetheless, effectively mitigating CH_4_ emissions without compromising animal production performance continues to pose a significant challenge in practical production. Consequently, future endeavors should encompass: firstly, a comprehensive exploration of the specific mechanisms by which feed additives and ingredients impact rumen microorganisms; secondly, ensuring the long-term viability of economic benefits and emission reduction effects in actual production settings; finally, synergistically employing multiple emission reduction strategies—such as selecting optimal feeds and utilizing effective additives—to maximize the emission reduction effect.

## 5. Conclusions

This study conducted a comprehensive analysis of relevant research papers in the field of rumen microbiota and CH_4_ emission spanning from 1998 to 2023. The analysis was performed using CiteSpace and VOSviewer software, providing a systematic review of the developmental trends within this field. The findings reveal an escalating trend in the number of pertinent publications, predominantly centered in Europe, America, and China. An analysis of keywords indicates a discernible shift in research focus from phenotypic indicators, such as growth performance and digestibility, towards mechanistic explorations of microorganisms, including protozoa and methanogens. The reduction in CH_4_ emissions from the gastrointestinal tract of ruminant livestock fundamentally hinges on encouraging hydrogen utilization within the rumen and preventing its use by methanogenic bacteria to produce CH_4_. Present nutritional regulation strategies encompass diet optimization, feed quality enhancement, increased rumen flow rate, addition of hydrogen pools, and methanogenic bacterial inhibition. However, the enduring effects of certain CH_4_ reduction techniques, such as oil and nitrate addition, require further investigation. Furthermore, pasture management and genetic breeding also serve as significant CH_4_ emission reduction tools. In a production context, it is crucial to consider both current production efficiency and potential CH_4_ reduction capabilities when devising a feasible mitigation plan. Future research should focus on synergizing different nutritional regulation strategies, assessing the sustainability of CH_4_ reduction outcomes, developing low-CH_4_-emitting livestock breeds, evaluating economic implications of livestock production systems, ensuring food safety, and addressing consumer preferences.

## Figures and Tables

**Figure 1 animals-15-00681-f001:**
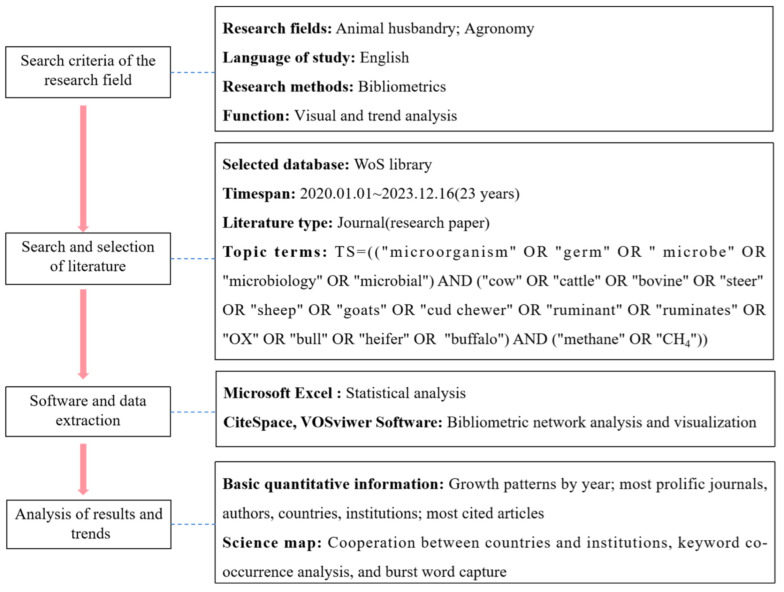
Flowchart of research procedure.

**Figure 2 animals-15-00681-f002:**
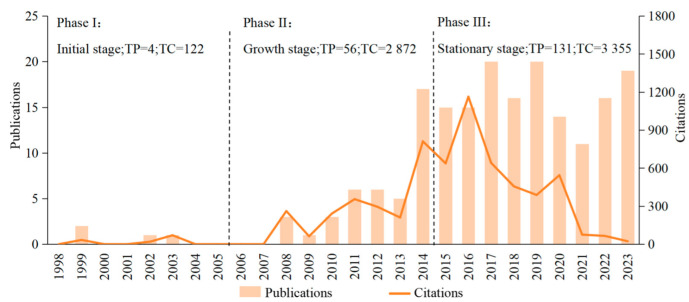
Number of publications and citations per year from 1998 to 2023 on Web of Science.

**Figure 3 animals-15-00681-f003:**
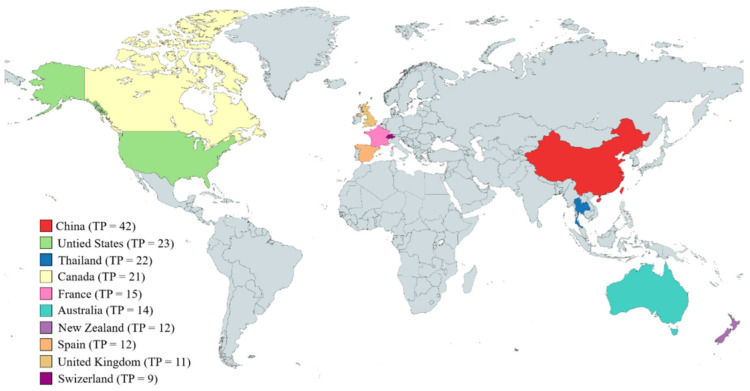
The number of publications for each geographic region in the world (based on the nationality of the first author).

**Figure 4 animals-15-00681-f004:**
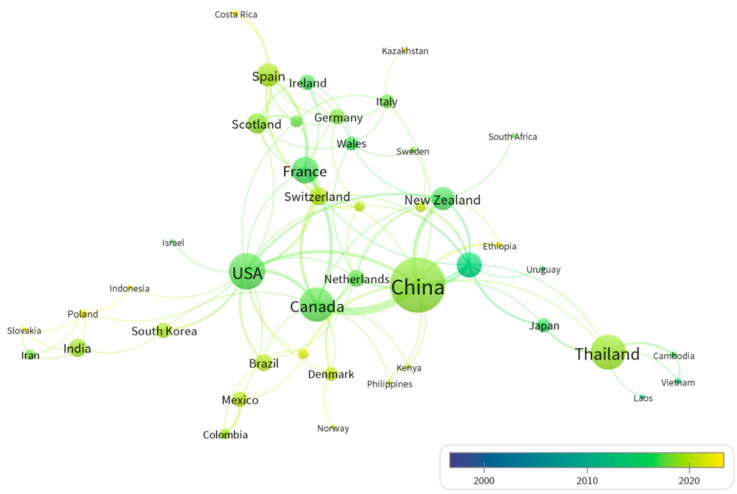
Research cooperation network among authors from different countries. Thicker lines indicate stronger partnerships.

**Figure 5 animals-15-00681-f005:**
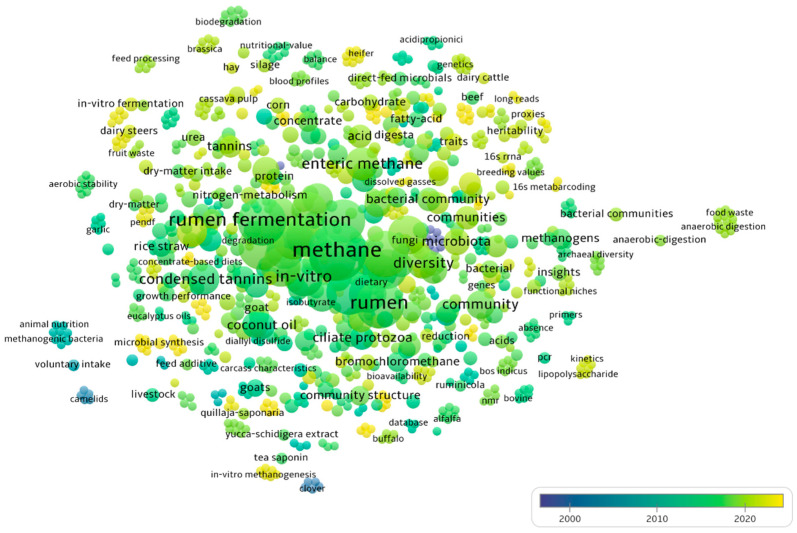
Research hotspots trends of change. Each small circle signifies a keyword. The size of the circle is directly proportional to the frequency of its occurrence. Furthermore, the thickness of the line connecting these circles indicates an increased co-occurrence frequency between keywords, signifying a closer relationship.

**Figure 6 animals-15-00681-f006:**
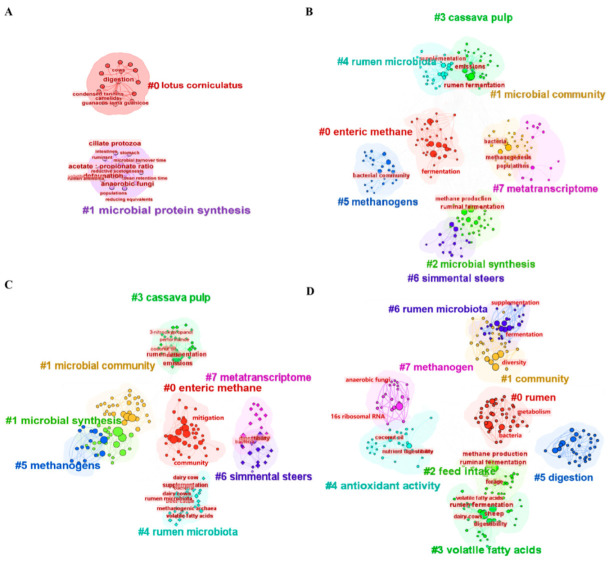
Keyword cluster analysis chart. (**A**) Phase I: 1998–2005; (**B**) phase II: 2006–2020; (**C**) phase III: 2021–2023; (**D**) all phases: 1998–2023.

**Figure 7 animals-15-00681-f007:**
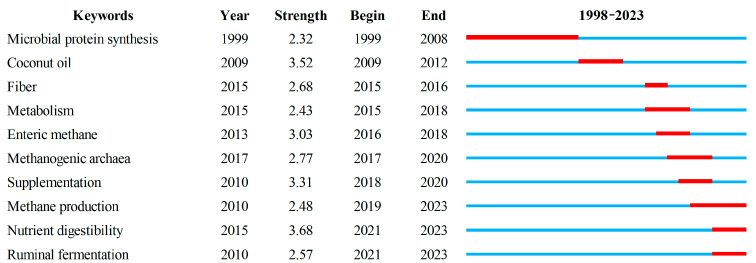
Top 25 keywords with the strongest citation bursts. Red lines indicate the years when the keyword was used frequently. Blue lines indicate the years when the keyword was not used frequently between 1998 and 2023.

**Table 1 animals-15-00681-t001:** The 10 most frequently published journals in the field of rumen microbiota and methane mitigation in ruminants from 1998 to 2023 with the relative number of articles.

No.	Journal	IF	Publications	Citations	Average Citations/Publication	JCR Categories
1	*Journal of Dairy Science*	3.5	19	1015	53.42	Dairy & Animal Science (Q1/9/62) Food Science & Technology (Q2/57/142)
2	*Frontiers in Microbiology*	5.2	19	613	32.26	Microbiology (Q2/38/135)
3	*Journal of Animal Science*	3.3	14	522	37.29	Dairy & Animal Science (Q1/10/62)
4	*Animal Feed Science and Technology*	3.2	13	454	34.92	Dairy & Animal Science (Q1/11/62)
5	*Asian-Australasian Journal of Animal Sciences*	2.7	8	159	19.88	Dairy & Animal Science (Q2/17/62)
6	*Animals*	3.0	8	117	14.63	Dairy & Animal Science (Q1/12/62) Veterinary Science (Q1/13/144)
7	*Journal of Animal Physiology and Animal Nutrition*	2.7	7	139	19.86	Dairy & Animal Science (Q1/13/62) Veterinary Science (Q1/19/144)
8	*Animal*	3.6	7	95	13.57	Dairy & Animal Science (Q1/7/62) Veterinary Science (Q1/9/144)
9	*Livestock Science*	1.8	6	253	42.17	Dairy & Animal Science (Q2/29/62)
10	*Animal Production Science*	2.2	6	76	12.67	Dairy & Animal Science (Q3/37/62)

**Table 2 animals-15-00681-t002:** Detailed information on the top 10 research publications about rumen microbiota and methane mitigation based on the total cited from 1998 to 2023.

No.	Title	Journal and Year	Average Citations per Year	Impact Factor and Category Quartile	Country	Total Cited	Ref.
1	Specific microbiome-dependent mechanisms underlie the energy harvest efficiency of ruminants	*ISME Journal*, 2016	51.13	11.0 (Q1)	Israel	409	[32]
2	The rumen microbial metagenome associated with high methane production in cattle	*BMC Genomics*, 2015	23.67	4.70 (Q1)	UK	213	[33]
3	Nitrate and sulfate: Effective alternative hydrogen sinks for mitigation of ruminal methane production in sheep	*Journal of Dairy Science*, 2010	14.57	3.50 (Q1)	The Netherlands	204	[34]
4	Bovine host genetic variation influences rumen microbial methane production with best selection criterion for low methane emitting and efficiently feed converting hosts based on metagenomic gene abundance	*PLOS Genetics*, 2016	24.88	4.50 (Q1)	UK	199	[35]
5	Two different bacterial community types are linked with the low-methane emission trait in sheep	*PLoS ONE*, 2014	18	3.70 (Q1)	New Zealand	180	[36]
6	Rumen metagenome and metatranscriptome analyses of low methane yield sheep reveals a Sharpea-enriched microbiome characterised by lactic acid formation and utilisation	*Microbiome*, 2016	21.75	15.5 (Q1)	New Zealand	174	[37]
7	Host genetics and the rumen microbiome jointly associate with methane emissions in dairy cows	*PLOS Genetics*, 2018	25.67	11.0 (Q1)	Denmark	154	[38]
8	Methane production in dairy cows correlates with rumen methanogenic and bacterial community structure	*Frontiers in Microbiology*, 2017	25.5	6.20 (Q2)	Sweden	153	[39]
9	Effects of addition of tea saponins and soybean oil on methane production, fermentation and microbial population in the rumen of growing lambs	*Livestock Science*, 2010	10.14	1.80 (Q2)	China	142	[18]
10	Methane emission by goats consuming diets with different levels of condensed tannins from lespedeza	*Animal Feed Science and Technology*, 2008	8.38	3.60 (Q1)	US.	134	[40]

## Data Availability

Dataset available on request from the authors.

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
