# Peer review of "Hotspot Analysis of Rumen Microbiota and Methane Mitigation in Ruminants: A Bibliometric Analysis from 1998 to 2023"

_animals, 2025, doi:10.3390/ani15050681_

Round 1
Reviewer 1 Report
Comments and Suggestions for Authors
The analysis of the published literature is mainly developed towards the country where the studies have been carried out, number of citations, collaborations, to mention a few points. It is the narrative of who does more, who is most cited, etc. However, this does not agree with the objective of offering a guide to direct future research, corresponding to point 4 of the results and discussion, which is poorly developed considering that it is the objective of the study.
Given the objective, the authors must analyze and communicate the advances achieved in microbiota research and methane mitigation. Thus, readers will have greater knowledge to detect the missing information to develop research on the subject.
Comments on the Quality of English Language
A thorough review of English is required, as well as correction of writing errors.
Author Response
Comments: The analysis of the published literature is mainly developed towards the country where the studies have been carried out, number of citations, collaborations, to mention a few points. It is the narrative of who does more, who is most cited, etc. However, this does not agree with the objective of offering a guide to direct future research, corresponding to point 4 of the results and discussion, which is poorly developed considering that it is the objective of the study. Given the objective, the authors must analyze and communicate the advances achieved in microbiota research and methane mitigation. Thus, readers will have greater knowledge to detect the missing information to develop research on the subject.
Response: We greatly appreciate your suggestion. Current research primarily focuses on short-term animal feeding experiments, usually spanning 21 to 42 days, with a noticeable lack of studies on long-term strategies. Different direct measurement methods such as respiration chambers, sulfur hexafluoride tracing technology, and the GreenFeed system are used to evaluate emission reduction, resulting in some degree of uncertainty and inconsistency in assessing CH4 emissions reduction. To mitigate this issue, future research should explore the differences between these techniques to improve their accuracy and data credibility. It is also crucial to identify and selectively regulate key microorganisms in low-methane-emitting cattle and sheep to reduce methane emissions. Moreover, individual emission reduction strategies often show limited efficiency; hence, future research should ensure these strategies do not negatively impact animal production performance and product quality. The formulation of multi-combination emission reduction strategies will be an effective way to reduce methane emissions from ruminants. The manuscript has been comprehensively revised to address these points. We hope the manuscript has been revised satisfactorily and our responses meet your expectations (Line 369-420).
Comments:A thorough review of English is required, as well as correction of writing errors.
Response: We have collaborated with international experts to refine the manuscript and aim to meet your expectations. If further modifications are needed, please let us know.
Reviewer 2 Report
Comments and Suggestions for Authors
Dear author!
Dear Editor,
The article is relevant and well written, the only remark on the design of the table should be included in the text.
The methodology describes very strongly how good the Was system is, which seems excessive to me.
The article is an overview, not an exploratory one.
It is recommended to accept it after minor edits.
Author Response
Comments: The article is relevant and well written, the only remark on the design of the table should be included in the text.
Response: We agree with your suggestion and add to it in the manuscript (Line 190).
Comments: The methodology describes very strongly how good the Was system is, which seems excessive to me.
Response: This study primarily utilizes the bibliometric method, which elucidates the dynamics and trends of disciplinary development. By conducting statistical analyses on indicators such as publication count, citation frequency, and author distribution, researchers can discern the prominent topics, emerging trends, and focal points within a specific discipline or research domain. Such insights enable scientists to stay at the forefront of research, allowing for timely adjustments to their research strategies and minimizing redundant efforts and resource wastage. Moreover, bibliometric articles offer an objective foundation for academic evaluation. Within the academic realm, a paper's quality and impact serve as critical metrics for assessing a researcher's scholarly stature. Bibliometrics, via quantitative analysis, can evaluate a paper's citation frequency and impact factor, thereby providing a framework for academic assessment and recognition. This aids in creating a more equitable and transparent academic evaluation system, incentivizing researchers to strive for excellence. Additionally, bibliometric articles can pinpoint gaps in academic research. A comprehensive analysis of literature data might reveal certain fields or topics that are under-explored or lack depth, highlighting areas of weakness in scholarly research. This assists researchers in identifying these gaps and offers guidance for future research endeavors. Lastly, bibliometric articles hold considerable relevance for research management and policy formulation. Through bibliometric analysis, one can gain insights into research resource allocation, institutional collaboration networks, and more, thereby supporting informed decision-making in research management and policy development. This facilitates the optimal distribution of research resources, encourages research collaboration and exchanges, and promotes interdisciplinary integration.
Comments: The article is an overview, not an exploratory one. It is recommended to accept it after minor edits.
Response: This paper primarily undertakes a review of the progress in research within this field, revealing that the majority of existing studies focus on short-term animal feeding experiments, with a relative paucity of research dedicated to long-term strategies. Additionally, the evaluation of CH4 emission reduction effects is often achieved through various direct measurement techniques such as respiration chambers, sulfur hexafluoride tracing technology, and the GreenFeed system. This leads to a certain degree of uncertainty and inconsistency in the evaluation of CH4 emission reduction. To address this issue, we propose that future research should delve deeply into exploring the disparities between different measurement techniques to enhance the accuracy and credibility of data. Moreover, identifying and selectively regulating key microorganisms in the rumen of low-CH4-emitting cattle and sheep to reduce CH4 emissions continues to be a research priority. Finally, singular CH4 emission reduction strategies tend to demonstrate limited efficiency in reducing CH4 emissions. Future research should ensure that these strategies do not negatively impact animal production performance and product quality. Conducting research on multiple, combined emission reduction strategies will be an important approach to effectively reducing CH4 emissions from ruminants.
Reviewer 3 Report
Comments and Suggestions for Authors
The manuscript Hotspot analysis of rumen microbiota and methane mitigation 1 in ruminants: A bibliometrics analysis from 1998-2023 is clear and generally well-written.
This study conducted a comprehensive analysis of relevant research papers in the field of rumen microbiota and CH4 emissions, covering the period from 1998 to 2023. The analysis was performed using Citespace and VOSviewer software to provide a systematic review of development trends in the field. In addition, a bibliometric analysis of highly cited articles, high-yielding countries or institutions, key journals in the field and keywords was carried out.
The work is well-founded, although some minor suggestions can be made (listed below). However, from a scientific point of view, it is not particularly interesting to know which country has contributed the most, or the co-authorship between countries, it seems more a bibliographic issue than animal husbandry. Much more emphasis should be placed on the topics of the papers and the contribution to general knowledge on methane mitigation strategies. Very few papers are described in detail and few more references could be included.
We agree with the authors that there is still a significant lack of effective long-term CH4 mitigation strategies.
Line 111: check dates
Line 176-181: repeated paragraph
Line 303: add references
Line 402: methanosarcina check the spelling
Line 405-410: this a repetition of the MM section
Author Response
Comments: The manuscript“Hotspot analysis of rumen microbiota and methane mitigation in ruminants: A bibliometrics analysis from 1998-2023”is clear and generally well-written. This study conducted a comprehensive analysis of relevant research papers in the field of rumen microbiota and CH4 emissions, covering the period from 1998 to 2023. The analysis was performed using Citespace and VOSviewer software to provide a systematic review of development trends in the field. In addition, a bibliometric analysis of highly cited articles, high-yielding countries or institutions, key journals in the field and keywords was carried out. The work is well-founded, although some minor suggestions can be made (listed below). However, from a scientific point of view, it is not particularly interesting to know which country has contributed the most, or the co-authorship between countries, it seems more a bibliographic issue than animal husbandry. Much more emphasis should be placed on the topics of the papers and the contribution to general knowledge on methane mitigation strategies. Very few papers are described in detail and few more references could be included. We agree with the authors that there is still a significant lack of effective long-term CH4 mitigation strategies.
Response: This study primarily employs bibliometrics to examine the quantity of publications, citation frequency, and author distribution currently associated with rumen microorganisms and methane emissions. This analysis aids in identifying the trends and focal points within the discipline, thereby enabling researchers to remain current, circumvent redundancy, and enhance their strategies effectively. The evaluation of citation frequency and impact factor also provides an objective foundation for academic assessment, thereby fostering fairness and transparency. Our research reveals that the majority of extant studies within this field are short-term animal feeding experiments (lasting between 21 and 42 days), while long-term strategy research is relatively limited. We also find that a variety of direct measurement methods, including respiration chambers, sulfur hexafluoride tracing technology, and the GreenFeed system, are employed to evaluate the emission reduction effect. This leads to a certain degree of uncertainty and inconsistency when assessing the CH4 emission reduction effect. To address this issue, we suggest future research should delve deeper into exploring the differences between various measurement techniques to enhance measurement accuracy and data reliability. The focus of upcoming research will continue to be the exploration of low methane-emitting cattle and sheep rumen key microorganisms and the implementation of targeted regulation to reduce methane emissions. However, a single emission reduction strategy often demonstrates low methane reduction efficiency. Therefore, future research must ensure that it does not adversely affect animal production performance and product quality. Concurrently, research into multiple combined emission reduction strategies will emerge as an important approach to reducing methane emissions from ruminants (Line 370-420).
Line 111: check dates
Response: We have reviewed and revised this dates and have included appropriate references. We welcome any further suggestions for improvement (Line 117-118).
Line 176-181: repeated paragraph
Response: We regret the mistake and have removed the sentence in the manuscript.
Line 303: add references
Response: Here is the description for the results in Figure 5, which we have revised in the manuscript (Line 294-297).
Line 402: methanosarcina check the spelling
Response: We have rewritten this section.
Line 405-410: this a repetition of the MM section
Response: We have rewritten this section (Line 370-397).
Reviewer 4 Report
Comments and Suggestions for Authors
The study aimed to provide a comprehensive overview of current publications about rumen microbiota and methane mitigation. This research is valuable in offering scientific guidance for future research on this important theme.
Data sources and research strategies were explained in detail what is important to verify the quality of the project.
Minor Comments
Line 10: Abstract
Line 107: Fig. to Figure 1. Leave the writing the same in the manuscript.
Major Comments
Line 402 - 403: Future efforts could potentially achieve effective CH4 mitigation by targeting specific microbial groups within the CH4 production pathway. Please cite articles and possibilities for achieving your suggestion for future efforts.
The conclusion of an article should summarize the main ideas discussed throughout the text and reinforce the objective of the study. In addition, it should present the implications of the work. Please rewrite.
If the Research patterns are predominantly concentrated in developed nations, why cite other countries? I recommend removing the sentence after Zhang et al. (2022) [58] until line 385 or including these countries in the research to offer a concise background for discussion.
Author Response
The study aimed to provide a comprehensive overview of current publications about rumen microbiota and methane mitigation. This research is valuable in offering scientific guidance for future research on this important theme. Data sources and research strategies were explained in detail what is important to verify the quality of the project.
Response: We appreciate your positive valuation of our works, which has encouraged us a lot. We have tried our best to revise the MS, so that all reviewers could like the contribution of this work.
Minor Comments
Line 10: Abstract
Response: We apologize for the oversight, and have accordingly made revisions in Line 23.
Line 107: Fig. to Figure 1. Leave the writing the same in the manuscript.
Response: We have modified it to "Figure..." in compliance with the journal's format, as shown in Line 113, 296-297.
Major Comments
Comments: Line 402 - 403: Future efforts could potentially achieve effective CH4 mitigation by targeting specific microbial groups within the CH4 production pathway. Please cite articles and possibilities for achieving your suggestion for future efforts.
Response: We greatly appreciate your suggestion. Current research primarily focuses on short-term animal feeding experiments, usually spanning 21 to 42 days, with a noticeable lack of studies on long-term strategies. Different direct measurement methods such as respiration chambers, sulfur hexafluoride tracing technology, and the GreenFeed system are used to evaluate emission reduction, resulting in some degree of uncertainty and inconsistency in assessing CH4 emissions reduction. To mitigate this issue, future research should explore the differences between these techniques to improve their accuracy and data credibility. It is also crucial to identify and selectively regulate key microorganisms in low-methane-emitting cattle and sheep to reduce methane emissions. Moreover, individual emission reduction strategies often show limited efficiency; hence, future research should ensure these strategies do not negatively impact animal production performance and product quality. The formulation of multi-combination emission reduction strategies will be an effective way to reduce methane emissions from ruminants. The manuscript has been comprehensively revised to address these points. We hope the manuscript has been revised satisfactorily and our responses meet your expectations (Line 370-397).
Comments: The conclusion of an article should summarize the main ideas discussed throughout the text and reinforce the objective of the study. In addition, it should present the implications of the work. Please rewrite.
Response: This study conducted a comprehensive analysis of relevant research papers in the field of rumen microbiota and CH4 emission, spanning from 1998 to 2023. The analysis was performed using Citespace and VOSviewer software, providing a systematic review of the developmental trends within this field. The findings reveal an escalating trend in the number of pertinent publications, predominantly centered in Europe, America, and China. An analysis of keywords indicates a discernible shift in research focus from phenotypic indicators, such as growth performance and digestibility, towards mechanistic explorations of microorganisms including protozoa and methanogens. The reduction of methane emissions from the gastrointestinal tract of ruminant livestock fundamentally hinges on encouraging hydrogen utilization within the rumen and preventing its use by methanogenic bacteria to produce methane. Present nutritional regulation strategies encompass diet optimization, feed quality enhancement, increased rumen flow rate, addition of hydrogen pools, and methanogenic bacterial inhibition. However, the enduring effects of certain methane reduction techniques, such as oil and nitrate addition, require further investigation. Furthermore, pasture management and genetic breeding also serve as significant methane emission reduction tools. In a production context, it is crucial to consider both current production efficiency and potential methane reduction capabilities when devising a feasible mitigation plan. Future research should focus on synergizing different nutritional regulation strategies, assessing the sustainability of methane reduction outcomes, developing low-methane-emitting livestock breeds, evaluating economic implications of livestock production systems, ensuring food safety, and addressing consumer preferences. We hope that the conclusion has been revised appropriately and our responses are satisfactory (Line 399-420).
Comments: If the Research patterns are predominantly concentrated in developed nations, why cite other countries? I recommend removing the sentence after Zhang et al. (2022) [58] until line 385 or including these countries in the research to offer a concise background for discussion.
Response: We agree with your suggestion and have deleted this sentence.
Round 2
Reviewer 1 Report
Comments and Suggestions for Authors
The authors have improved the work presented previously, they have focused their efforts on highlighting the topic of scientific literature generated from 1998 to 2023 on ruminal microbiota and methane mitigation.
Comments on the Quality of English Language
A specialized review is required.
Author Response
Comments: The authors have improved the work presented previously, they have focused their efforts on highlighting the topic of scientific literature generated from 1998 to 2023 on ruminal microbiota and methane mitigation.
Response: Thank you for these positive comments of our revised manuscript.
Reviewer 4 Report
Comments and Suggestions for Authors
The corrections were addressed appropriately.
Author Response
Comments: The corrections were addressed appropriately.
Response: Thank you for these positive comments of our revised manuscript.